# The Role of Marek’s Disease Virus UL12 and UL29 in DNA Recombination and the Virus Lifecycle

**DOI:** 10.3390/v11020111

**Published:** 2019-01-28

**Authors:** Renato L. Previdelli, Luca D. Bertzbach, Darren J. Wight, Tereza Vychodil, Yu You, Sina Arndt, Benedikt B. Kaufer

**Affiliations:** Institute of Virology, Freie Universität Berlin, 14163 Berlin, Germany; previnato@zedat.fu-berlin.de (R.L.P.); luca.bertzbach@fu-berlin.de (L.D.B.); Tereza.Vychodil@fu-berlin.de (T.V.); yuyou@zedat.fu-berlin.de (Y.Y.); sinarndt@gmail.com (S.A.)

**Keywords:** herpesvirus, Marek’s disease virus, virus replication, latency, recombination

## Abstract

Marek’s disease virus (MDV) is an oncogenic alphaherpesvirus that infects chickens and integrates its genome into the telomeres of latently infected cells. MDV encodes two proteins, UL12 and UL29 (ICP8), that are conserved among herpesviruses and could facilitate virus integration. The orthologues of UL12 and UL29 in herpes simplex virus 1 (HSV-1) possess exonuclease and single strand DNA-binding activity, respectively, and facilitate DNA recombination; however, the role of both proteins in the MDV lifecycle remains elusive. To determine if UL12 and/or UL29 are involved in virus replication, we abrogated their expression in the very virulent RB-1B strain. Abrogation of either UL12 or UL29 resulted in a severe impairment of virus replication. We also demonstrated that MDV UL12 can aid in single strand annealing DNA repair, using a well-established reporter cell line. Finally, we assessed the role of UL12 and UL29 in MDV integration and maintenance of the latent virus genome. We could demonstrate that knockdown of UL12 and UL29 does not interfere with the establishment or maintenance of latency. Our data therefore shed light on the role of MDV UL12 and UL29 in MDV replication, DNA repair, and maintenance of the latent virus genome.

## 1. Introduction

Marek’s disease virus (MDV), also known as Gallid herpesvirus 2 (GaHV-2), is an alphaherpesvirus that infects chickens and can cause a variety of clinical symptoms [1]. These include immunosuppression, neurological disorders and, most importantly, fatal T cell lymphomas [2]. MDV is prevalent worldwide and induces the most frequent clinically diagnosed cancer in animals [3]. During primary infection, MDV is taken up via the respiratory route, where it infects macrophages or dendritic cells. These cells transfer the virus to the major lymphoid organs, where it infects B and T cells [4]. The infection of T cell subsets plays a crucial role for MDV pathogenesis, while B cells are dispensable for this process [5]. CD4+ T cells are the predominant cells for the establishment of MDV latency, and can be transformed, resulting in T cell lymphomas. 

We and others recently demonstrated that MDV integrates its genome into the telomeres of latently infected cells and tumor cells [6,7,8,9,10]. This integration has also been shown for other herpesviruses, including human herpesvirus 6A (HHV-6A), 6B (HHV-6A), and 7 (HHV-7) [11,12]. These herpesviruses and several others harbor telomeric (TTAGGG)_n_ repeat arrays (TMR), identical to host telomere sequence at the end of their virus genomes [8,13]. We have previously established that these TMR arrays facilitate integration of MDV and HHV-6A [6,7,14]. The MDV genome harbors two distinct TMR arrays, the multiple TMR (mTMR) with up to 100 repeats in length, and the short TMR (sTMR) with a fixed number of six repeats [6]. Both types of TMR have been shown to be crucial for efficient MDV integration [6,7]. While the sequences that facilitate MDV integration are well characterized, the proteins that facilitate this process remain elusive.

MDV encodes two proteins, called UL12 and UL29, that are conserved among herpesviruses and have been shown to facilitate recombination during herpes simplex virus 1 (HSV-1) replication [15], a process that is not fully understood. For HSV-1, the UL12 protein has been shown to possess 5′-3′ exonuclease activity [16,17]. UL12 has also been shown to enhance DNA repair through single strand annealing (SSA), a process that could potentially facilitate MDV integration. UL29 is also known as infected cell protein 8 (ICP8) and has single strand DNA-binding activity [15,17,18], a function that plays a crucial role in many recombination processes. HSV-1 UL12 and UL29 interact with each other and facilitate strand exchange in vitro, which could conceivably initiate MDV integration. Even though UL12 and UL29 are present in all members of the *Herpesviridae* family, they are not highly conserved on the sequence level. UL12 and UL29 of MDV and HSV-1 only have a sequence homology of 36.2% and 43.7% respectively, suggesting that they may not have the same functions for MDV. While both genes are well characterized for HSV-1, the roles of the MDV orthologues in replication, recombination, and integration remains unknown.

To determine the role of UL12 and UL29 in the MDV lifecycle, we generated recombinant viruses that do not express the respective proteins. We could demonstrate that abrogation of UL12 and UL29 either severely impairs or abolishes MDV replication. In addition, we could show that MDV UL12 aids in SSA DNA repair, a process that could facilitate MDV integration. Finally, our experiments revealed that UL12 and UL29 are dispensable for MDV integration and the maintenance of the latent virus genome, indicating that these two proteins either do not facilitate integration or that other proteins complement for the loss of these proteins.

## 2. Materials and Methods

### 2.1. Cells and Viruses

Chicken embryo cells (CEC) were obtained from Valo Biomedia specific pathogen-free (SPF) embryos, as described previously [19]. Mutant viruses were reconstituted by transfection of CEC with purified bacterial artificial chromosome (BAC) DNA by calcium phosphate transfection [20]. All reconstituted viruses were amplified on CEC for 2 to 4 passages and infected CEC were stored in liquid nitrogen. To confirm the presence of the introduced mutations, viral DNA was extracted from infected CEC using an RTP DNA/RNA Virus Mini Kit (STRATEC, Germany), and the target region was analyzed by DNA sequencing using the respective primers (Appendix A). In addition, we used a previously published virus that lacks the mTMR sequences (∆mTMR) and is defective in integration into host telomeres [5,6]. Reticuloendotheliosis virus-transformed T cells (CU91) were propagated in RPMI 1640 media (PAN Biotech, Germany) supplemented with 1% sodium pyruvate, 1% nonessential amino acids, 10% FBS, and penicillin–streptomycin, and maintained at 41°C in a 5% CO_2_ atmosphere [21,22]. *Cairina* retina (CR) cells [23] were maintained in Dulbecco’s MEM/Ham’s F-12 (1:1, Biochrom, Germany), 5% FBS, and penicillin–streptomycin. Double strand break repair (DSBR) reporter cell lines HEK293 DR-GFP, SA-GFP, EJ2-GFP, and U2OS-EJ5-GFP were cultured in DMEM (PAN Biotech, Germany) supplemented with 10% FBS, penicillin–streptomycin, and were kindly provided by Dr. Sandra Weller (University of Connecticut Health Center) [19,22].

### 2.2. Generation of Recombinant Viruses

The following recombinant viruses were generated based on pRB-1B, an infectious BAC clone of the highly oncogenic RB-1B strain, using two-step Red-mediated mutagenesis as previously described [20]. To abrogate expression of the full-length UL12 protein, we replaced amino acid (aa) 6 and 7 with two stop codons (UL12 mut1) (Figure 1A). Since a second shorter C-terminal form of HSV-1 UL12 has been described previously [24], we also inserted a stop codon at aa 136 into UL12 mut1 or wild type, resulting in UL12 mut2 and UL12 mut3, respectively. In the case of UL29, the start codon was replaced by a stop codon to abrogate its expression (UL29 mut) (Figure 1A). In addition, revertant viruses were generated for all recombinant viruses, in which the original sequence was restored to confirm that the observed phenotypes are due to the introduced mutations. Primers used for the generation of these recombinant viruses are listed in Appendix A. Final clones were confirmed by multiple restriction fragment length polymorphism (RFLP) analyses, PCR, and Sanger sequencing of the target region [7].

### 2.3. Plaque Size Assays and Growth Kinetics

Replication properties of recombinant viruses were initially determined by plaque size assays, as previously described [7,25]. In case the introduced mutations abrogated MDV replication, plaque sizes were measured after transfection of the viral BAC DNA. Briefly, images of 50 randomly selected plaques per well were taken, and plaque areas determined using the ImageJ software (NIH) and were normalized to wild type virus. Replication properties were confirmed by multistep growth kinetics, as previously described [26].

### 2.4. DNA Repair Assays

To determine if UL12 or UL29 activate double strand break repair (DSBR) pathways, we used the four well-established reporter cell lines HEK293 DR-GFP, SA-GFP, EJ2-GFP, and U2OS EJ5-GFP. They allowed us to monitor repair of an I-*Sce*I generated double strand break (DSB) by i) homologous recombination (HR), ii) single strand annealing SSA, iii) total non-homologous end joining (NHEJ) or iv) alternative NHEJ (A-NHEJ), respectively [17,27,28]. To assess the role of UL12 and UL29 in these repair assays, we cloned UL12 of the RB-1B strain into pcDNA3.1+. Due to the previously observed toxicity of the UL29 in *E.coli* [29], we used the low copy pHA vector for the expression of UL29. These expression vectors were used for the DNA repair assay, as previously described [17]. Briefly, HEK293 HRGFP, SA-GFP, EJ2-GFP, or U2OS EJ5-GFP SA-GFP cells were plated in 12 well plates treated with 0.01% poly-L-lysine (Sigma-Aldrich, country). The next day, cells were transfected with the control vectors, the UL12 or UL29 plasmid in combination with the I-*Sce*I expression vector (pCBA-SceI, Addgene: #26477), and an E2-Crimson transfection control (pE2-Crimson, Clontech, USA) using Lipofectamine2000, according to the manufacturer’s instructions (Invitrogen, country). Cells were analyzed by flow cytometry 72 hours post-transfection, as previously described [29].

### 2.5. In Vitro Latency Assay

One million CEC were infected with 30,000 plaque forming units (PFU) of cell-associated virus. Four days post-infection, infected CEC were overlaid with one million CU91 T cells. After 16 h, CU91 T cells were collected and the cells passaged for 14 days. Samples were collected on day 1 and 14, and the virus genome quantified by qPCR. MDV genome copies were determined using specific primers, and a probe for the viral ICP4 gene that was normalized against the cellular inducible nitric oxide synthase (iNOS) gene, as previously described [5].

### 2.6. shRNA Knockdown of UL12 and UL29

To confirm our in vitro latency assay data for UL12 mut1, we used shRNAs targeting the 5’- and 3’-ends of UL12 and UL29 mRNA. The two shRNAs for each gene were cloned into the pLKO5.shRNA vectors, lentiviruses generated and delivered into CR and CU91 T cells as previously described [29]. CU91 T cells were selected using hygromycin (UL12) or puromycin (UL29), clonal lines were generated, and shRNA-expressing clones used for the in vitro latency assay. 

### 2.7. Statistical Analyses

Statistical analyses were performed using GraphPad Prism, version 7. Datasets were initially tested for normal distribution. Plaque size assay data were converted into plaque diameters and analyzed using one-way analysis of variance (ANOVA). Growth kinetics data were analyzed using Kruskal–Wallis and Mann–Whitney *U* tests. Data of the DNA repair assays were analyzed using the Mann–Whitney *U* test. Latency assays were analyzed using Kruskal–Wallis tests.

## 3. Results

### 3.1. Role of UL12 and UL29 in MDV Lytic Replication

To determine the role of UL12 in MDV replication, we replaced codons aa 6 and 7 with stop codons (UL12 mut1, Figure 1A), without affecting the overlapping coding sequence of UL13. Upon reconstitution of UL12 mut1, we performed plaque size assays and could demonstrate that abrogation of full-length UL12 significantly impaired virus replication by more than 30% compared to wild type and the revertant viruses (Figure 1B). We also confirmed this growth defect using multistep growth kinetics (Figure 1C). Since HSV-1 also encodes a shorter isoform of UL12, termed UL12.5, that localizes to the mitochondria [16,24], we mutated the second start codon at aa 136 (UL12 mut2, Figure 1A) to a stop codon, to abrogate this putative protein in MDV. Intriguingly, UL12 mut2 did not replicate upon virus reconstitution. Only single infected cells were detected after transfection of the virus BAC DNA (Figure 1D), while the revertant behaved comparably to wild type virus. To determine if insertion of the stop codon at aa 136 alone abrogates virus replication, we generated a mutant virus that only harbors this mutation (UL12 mut3). MDV replication was completely abrogated by insertion of this stop codon (Figure 1E), indicating that the C-terminus of UL12, including its putative isoform, is essential. To investigate the role of UL29 in MDV replication, we replaced the start codon of UL29 with a stop codon (UL29 mut, Figure 1A). Insertion of this stop codon completely abrogated MDV replication (Figure 2), highlighting that UL29 is also essential for MDV replication. Taken together, our data demonstrate that UL12 and UL29 play a crucial role in MDV lytic replication.

### 3.2. Role of UL12 and UL29 in Double Strand Break Repair

It has previously been shown that UL12 and UL29 of HSV-1 form a complex and possess strand exchange activity [15,17]. In addition, HSV-1 UL12 can aid in DNA repair via the single strand annealing (SSA) pathway [17]. To determine if UL12 and/or UL29 from MDV can aid in the repair of double strand DNA (dsDNA) breaks via homology repair pathways, we utilized well-established DNA repair reporter cell lines [29]. Each of the four reporter cell lines (HR, SSA, A-NHEJ, and NHEJ) were transfected with an I-*Sce*I expression plasmid in combination with plasmids expressing UL12 and/or UL29. Intriguingly, MDV UL12 increased SSA activity by approximately 3-fold when compared to the control cells (Figure 3A), while no effect was observed for the HR, A-NHEJ, and NHEJ pathways (Figure 3B–D). UL29 neither augmented the activity of UL12 nor enhanced any of the other pathways. These results are consistent with the observations made for HSV-1 UL12 and UL29, and suggest that MDV UL12 could aid virus integration via an SSA pathway.

### 3.3. Role of UL12 and UL29 MDV Integration and the Maintenance of the Latent Virus Genome

In the case of HSV-1, only the full-length UL12 protein localizes to the nucleus, where it aids in recombination [24]. In silico predictions confirmed that the nuclear localization signal (NLS) of MDV UL12 is located within the first 50 aa, as previously shown for HSV-1 [30]. To determine if UL12 plays a role in integration and genome maintenance, we used the UL12 mut1 virus that lacks the full-length UL12 protein and the NLS required for nuclear localization. CU91 T cells were infected with wild type virus, UL12 mut1, and a mTMR deletion mutant (∆mTMR) which has previously been shown to be defective in telomere integration [6]. Upon infection, cells were cultured for 14 days to determine the level of virus genome maintenance within the infected culture over time. Equal virus genome copies were detected after one day post-infection (Figure 4). After passaging, the ∆mTMR genome was almost completely lost, due to its inability to integrate into host telomeres, as previously shown [6]. In stark contrast, no effect was observed for UL12 mut1 compared to the wild type virus, indicating that the full-length UL12 (containing the NLS) is dispensable for MDV integration and genome maintenance. 

To confirm this observation, we generated CU91 T cells expressing shRNAs against UL12 or UL29 mRNAs that we validated in CR cells (Appendix A). Two cell lines for each viral target were generated that express two shRNAs each, one targeting the 5’-end and one the 3’-end of UL12 or UL29 mRNA. We used these cell lines to assess if virus genome maintenance and, therefore, integration is impaired. Upon infection, comparable virus genome copy levels were observed. As mentioned above, the ∆mTMR genome was almost completely lost due to an impairment in integration. Interestingly, shRNAs against UL12 or UL29 had no effect on virus genome maintenance, suggesting that UL12 and UL29 are not essential for MDV integration (Appendix A).

## 4. Discussion

Even though MDV has been studied for decades, the exact role of most viral genes in the MDV lifecycle remains elusive. Two of these genes are UL12 and UL29, whose HSV-1 orthologues encode a 5′–3′ exonuclease and a single-stranded DNA binding protein, respectively. These genes are involved in HSV-1 DNA replication and form a recombination complex that facilitates strand exchange in vitro [15,17,31]. Since the functions of MDV UL12 and UL29 remained elusive, we set out to investigate their roles in the MDV lifecycle.

We first determined if UL12 or UL29 are essential for MDV replication. We generated recombinant viruses with stop codon insertions in the coding sequence. Interestingly, both UL12 and UL29 play a crucial role in lytic replication of MDV, which is consistent with previous findings on their orthologues in HSV-1 [32,33]. While UL29 is essential for MDV replication, insertion of stop codons at the N-terminus of UL12 (UL12 mut1) resulted in viable virus. Replication of UL12 mut1 was severely impaired; however, we were able to generate virus stocks. The two stop codons are adjacent to the only predicted NLS in UL12 and would, therefore, abrogate the expression of the full-length protein, as previously shown for HSV-1 UL12 [30]. Insertion of a stop codon at aa 136 of UL12 completely abrogated MDV replication, suggesting that the C-terminus containing the predicted site of exonuclease activity is essential.

Next, we assessed whether UL12 and UL29 can aid in DNA damage repair using well-established reporter cell lines. MDV UL12 enhanced the repair of dsDNA breaks via the single strand annealing (SSA) pathway (Figure 3). This data is consistent with previous data on the orthologues of HSV-1 that showed UL12 aids in DNA repair via the SSA pathway [17]. By contrast, UL29 did not aid in SSA repair or augment the effect of UL12 (Figure 3), which is also consistent with recently published data on HHV-6A [29].

One aspect of the MDV lifecycle that sets it aside from most herpesviruses is the fact that it integrates its genome into the host telomeres [6,7]. We previously demonstrated that telomere arrays at the ends of the virus genome are crucial for the integration into host telomeres [6,7,14]. This suggests that integration occurs via a homology-directed repair between telomere repeats in the virus genome and telomeres of the host. As we could demonstrate that UL12 aids in the repair of homologous DNA sequences via SSA (Figure 3), we set out to assess the role of UL12 in MDV integration. First, we used UL12 mut1 harboring stop codons at aa 6 and 7 that abrogate expression of the full-length protein, including its NLS. We infected CU91 T cells and monitored maintenance of the virus genome over time. After infection with ∆mTMR, a mutant virus that has a defect in integration into host telomeres, the virus genome was rapidly lost (Figure 4). By contrast, the genome of UL12 mut1 was maintained at levels comparable to wild type virus, suggesting that integration was not affected. Even though the maintenance of MDV in these cells is stable over weeks, the number of cells was too low to be visually confirmed on a quantitative level, e.g., by fluorescence in situ hybridization (FISH). To confirm our results, we used specific shRNAs targeting UL12 or UL29 in CU91 T cells. Inhibition of UL12 or UL29 in chicken T cells did not affect genome maintenance (Appendix A), suggesting that these proteins are dispensable for MDV integration. Intriguingly, we addressed the role of the UL12 and UL29 orthologues of human herpesvirus 6A (HHV-6A) in parallel, and could demonstrate that these proteins are also not essential for HHV-6A integration [29].

Taken together, our results show that UL12 and UL29 play a major role in MDV replication in vitro. Surprisingly, even though UL12 aids in SSA DNA repair, a pathway that could potentially mediate MDV integration into host telomeres, it is dispensable for the maintenance of the integrated virus genome.

## Figures and Tables

**Figure 1 viruses-11-00111-f001:**
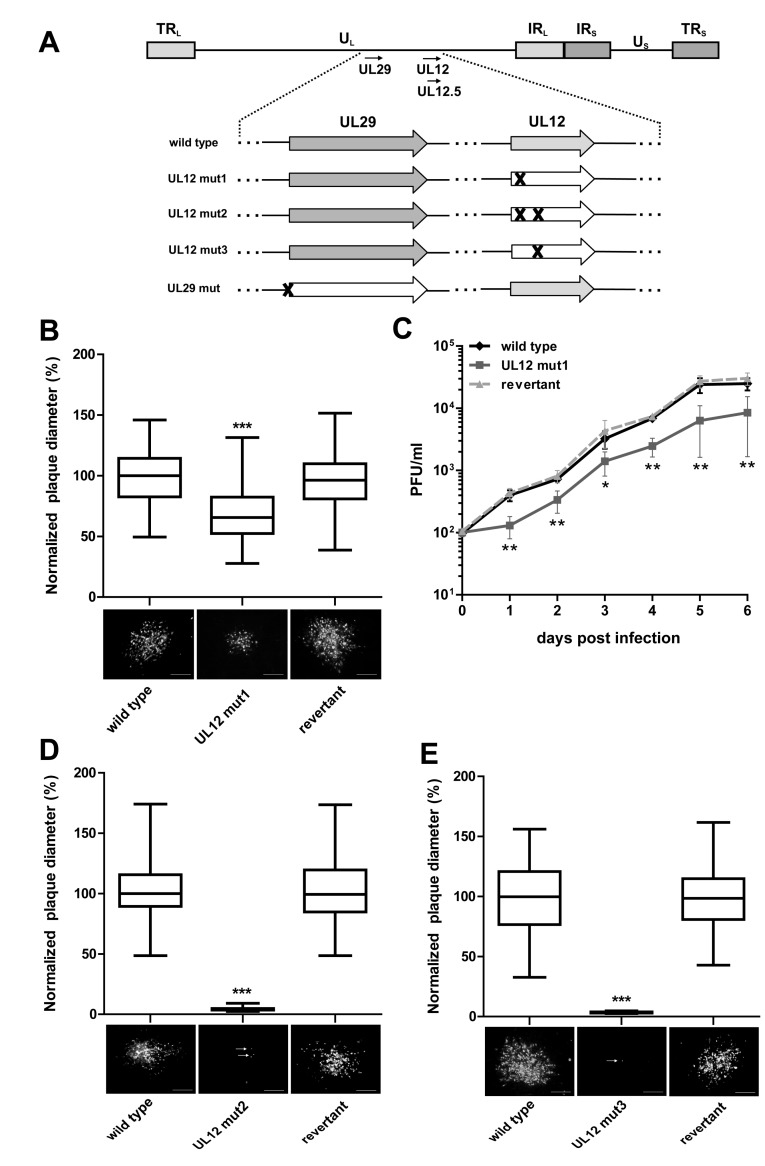
Generation and characterization of recombinant mutant viruses. (**A**) Schematic representation of the Marek’s disease virus (MDV) genome, with a focus on the unique long (UL) region containing the UL12 and UL29 genes. Mutations in UL12 (stop codon insertion at aa 6/7 and/or aa 136) and UL29 (stop codon at aa 1) are indicated. (**B**) Chicken embryo cells (CEC) were infected with 100 PFU of wild type MDV, UL12 mut1, or the revertant viruses. The average plaque diameters from three independent experiments, normalized to wild type, are shown in the box plots. Representative images of plaques are shown below (Scale bar, 100 µm). (**C**) CEC were infected with 100 PFU of wild type, UL12 mut1, and revertant viruses, and the titer determined at indicated times post-infection. Shown are the mean titers of these viruses from three independent experiments ± SEM (**, *p* ≤ 0.05; *, *p* = 0.01; Mann–Whitney *U* test). Wild type, (**D**) UL12 mut2, (**E**) UL12 mut3, and respective revertant MDV BACs were transfected into CEC and images of plaques acquired six days post-transfection. The average plaque diameters from three independent experiments are shown in box plots (normalized to wild type). Representative images of plaques are shown below (Scale bar, 100 µm). White arrows show single infected cells. Statistical differences in plaque diameters were determined using one-way ANOVA analyses (***, *p* < 0.0001).

**Figure 2 viruses-11-00111-f002:**
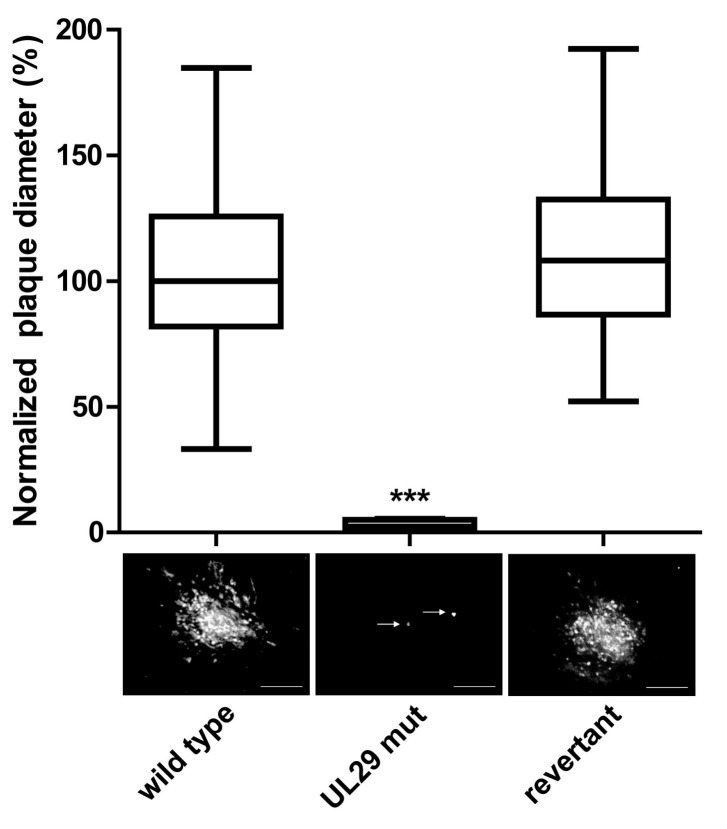
Characterization of the UL29 mutant virus. Wild type, UL29 mut, and revertant BACs were transfected into CEC cells and images of plaques acquired six days later. The average plaque diameters from three independent experiments are shown in a box plot (normalized to wild type) Representative images of plaques are shown below (scale bar, 100 µm). White arrows indicate single infected cells. Statistical differences in plaque diameters were determined using one-way ANOVA analyses (***, *p* < 0.0001).

**Figure 3 viruses-11-00111-f003:**
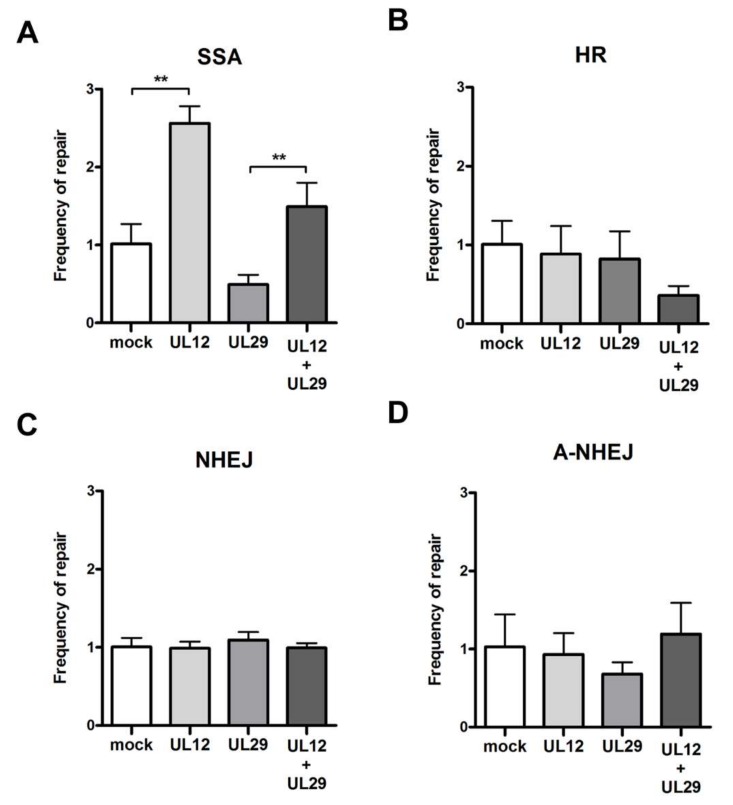
Role of UL12 and UL29 in DNA damage repair. Four DNA damage reporter cell lines were utilized, containing an integrated DNA damage reporter that expresses GFP when an induced dsDNA break is repaired by (**A**) single strand annealing (SSA), (**B**) homologous recombination (HR), (**C**) non-homologous end joining (NHEJ), and (**D**) alternative NHEJ (A-NHEJ). These cell lines were transfected with expression plasmids for UL12 and/or UL29, to assess whether these MDV proteins can aid in the specific DNA repair pathways. The mean frequency of repair for the indicated expression constructs is shown from six independent experiments ± SEM. Statistics were performed using the Mann–Whitney *U* test **, *p* < 0.05.

**Figure 4 viruses-11-00111-f004:**
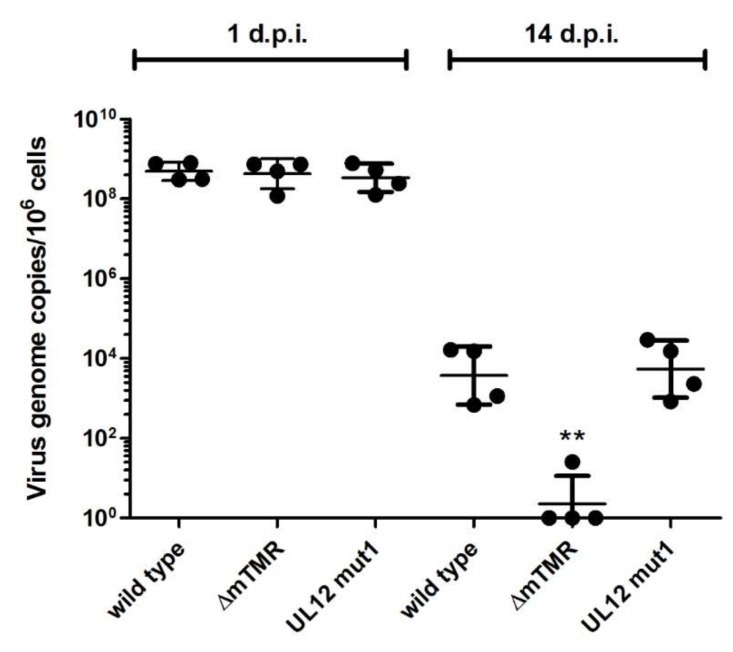
Viral genome maintenance in CU91 T cells infected with UL12 mut1. CU91 T cells were infected with different MDV mutants and infected cells cultured for 14 days. Samples were taken for qPCR analysis on day 1 and day 14. The plots show the mean level of virus genome copies per million cells from four independent experiments (each experiment is depicted by a single dot). Statistics were performed using the Kruskal–Wallis test (***p* < 0,01). d.p.i: days post-infection.

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
