# Peer review of "The Role of Marek’s Disease Virus UL12 and UL29 in DNA Recombination and the Virus Lifecycle"

_viruses, 2019, doi:10.3390/v11020111_

Round 1

Reviewer 1 Report

This study investigated the roles of the UL12 and UL29 proteins of Marek’s disease herpesvirus (MDV) in virus replication, DNA repair and the establishment and maintenance of virus latency.  The roles of UL12 and UL29 genes are not known for MDV, but are known for the related human herpesvirus HSV-1.  Assuming some functional homology between the MDV and HSV-1 genes, the authors considered that UL12 and UL29 could be essential for lytic replication of MDV and for DNA recombination, which could be important for the integration of the MDV genome into the host chromosome telomeres, an event which is key for host cell transformation by MDV.

MDV is an economically-important oncogenic virus of chickens, but the mechanism of neoplastic transformation and the role of virus proteins in genome integration are not fully understood.  Further insights into the molecular mechanisms of virus replication, latency and transformation will assist in the future design of more effective recombinant vaccines or other molecular controls for both MDV and other herpesviruses.  Therefore, this is an important study and the results provide novel information.

The roles of MDV UL12 and UL29 have been addressed by constructing UL12-mutant and UL29-mutant MDVs and using cellular and molecular assays.  All experiments have been conducted thoroughly and using appropriate controls.  The Introduction gives an appropriate background to the study and is well-referenced.  Materials & Methods are clearly described.  The Results are, mostly (see below), clearly presented and appropriately analysed and interpreted.  The Discussion put the results in context and is appropriately referenced, and the conclusions drawn are appropriate for the results.

I have no major criticisms, just a short list of errors, missing details and minor points.

(1)    Lines 13, 14: Should read either ‘…the roles….remain elusive’ or ‘…the role….remains elusive’.

(2)    Line 103: Change ‘..a E2-Crimson transfection…’ to ‘…an E2-Crimson transfection…’

(3)    Lines 95-96: The information about the four reporter cell lines would be better placed in the Materials & Methods section ‘Cells and viruses’.

(4)    Line 131: Revertant viruses are mentioned for the first time in the Results section.  These viruses should be introduced in the Materials & Methods section ‘Cells and viruses’.

(5)    Figure 1A: The UL29 mutant is referred to as ‘vUL29 mut’ in the figure, but just ‘UL29 mut’ elsewhere in the manuscript.

(6)    Figure 1 components and legend: Figure 1 has five components (A, B, C, D and E).  However, the legend only mentions A, B, C and D and, as currently written, these do not correspond correctly to the lettered components of the Figure.  A, B, C , D and E seem to have been correctly referenced in the main body of the Results text, but please check this as well.

(7)    Figure 1: Please make it clear that dpi is ‘days post infection’.

(8)    Line 173: Should say Fig 3B-D (not Fig 3C-D) I think.

(9)    Line 173: Reword to read: ‘UL29 neither augmented the activity of UL12 nor enhanced any of the other…’

(10)Lines 181, 182: Should read either ‘…The mean frequency…is shown…’, or ‘The mean frequencies…are shown…’

(11)Line 190: The TMR deletion virus is mentioned for the first time in the Results section.  This virus should be introduced in the Materials & Methods section ‘Cells and viruses’.

(12)Line 191: Change to read ‘Upon infection, cells were cultured…’

(13)Line 199: Change ‘shRNA’ to ‘shRNAs’.

(14)Lines 198-205: Is this the data which is shown in Figure S1?  If so, please reference Figure S1 here.

(15)Line 241: Should refer to Fig. 3 (not Fig. 5).

(16)Line 244: Change to ‘After infection with…’

(17)Reference list: There are many letters that should be in upper case, but are in lower case.  For example, Line 280 (T-cell), Lines 290, 292, 294 (Marek’s), Line 299 (HHV-6), Line 313 (UL12), Lines 323, 324, 326, 227, 331 (MDV).  Please check the whole reference list and correct where required.

Author Response

We would like to thank the reviewers and editors for the evaluation of our manuscript. We have addressed the concerns of the reviewers and performed the necessary changes, which improved the manuscript compared to the previous version. You can find our direct responses to the reviewers’ comments below:

Reviewer 1 Comments & Revisions:

(1)    Lines 13, 14: Should read either ‘…the roles….remain elusive’ or ‘…the role….remains elusive’.

Thanks for picking up this typographical error. It has been corrected as suggested by the reviewer. 

(2)    Line 103: Change ‘..a E2-Crimson transfection…’ to ‘…an E2-Crimson transfection…’

The text was changed as suggested by the reviewer. 

(3)    Lines 95-96: The information about the four reporter cell lines would be better placed in the Materials & Methods section ‘Cells and viruses’.

The information on the four reporter cell lines has been moved to the ‘Cells and viruses’ paragraph of the Materials & Methods section.

(4)    Line 131: Revertant viruses are mentioned for the first time in the Results section.  These viruses should be introduced in the Materials & Methods section ‘Cells and viruses’.

The information on the revertant viruses was introduced in the Materials & Methods section as suggested by the reviewer.

(5)    Figure 1A: The UL29 mutant is referred to as ‘vUL29 mut’ in the figure, but just ‘UL29 mut’ elsewhere in the manuscript.

Thanks. We corrected this in the figure.

(6)    Figure 1 components and legend: Figure 1 has five components (A, B, C, D and E).  However, the legend only mentions A, B, C and D and, as currently written, these do not correspond correctly to the lettered components of the Figure.  A, B, C , D and E seem to have been correctly referenced in the main body of the Results text, but please check this as well.

Thank you for picking this up. We corrected the figure legend accordingly.

(7)    Figure 1: Please make it clear that dpi is ‘days post infection’.

Done.

(8)    Line 173: Should say Fig 3B-D (not Fig 3C-D) I think.

Corrected.

(9)    Line 173: Reword to read: ‘UL29 neither augmented the activity of UL12 nor enhanced any of the other…’

Thank you for the suggestion. We changed the sentence as suggested by the reviewer.

(10)Lines 181, 182: Should read either ‘…The mean frequency…is shown…’, or ‘The mean frequencies…are shown…’

Corrected.

(11)Line 190: The TMR deletion virus is mentioned for the first time in the Results section.  This virus should be introduced in the Materials & Methods section ‘Cells and viruses’.

We agree and inserted the information in the Materials & Methods section.

(12)Line 191: Change to read ‘Upon infection, cells were cultured…’

Corrected.

(13)Line 199: Change ‘shRNA’ to ‘shRNAs’.

Corrected.

(14)Lines 198-205: Is this the data which is shown in Figure S1?  If so, please reference Figure S1 here.

Yes. We included the reference of Figure S2 (previously S1).

(15)Line 241: Should refer to Fig. 3 (not Fig. 5).

Corrected.

(16)Line 244: Change to ‘After infection with…’

Corrected.

(17) Reference list: There are many letters that should be in upper case, but are in lower case.  For example, Line 280 (T-cell), Lines 290, 292, 294 (Marek’s), Line 299 (HHV-6), Line 313 (UL12), Lines 323, 324, 326, 227, 331 (MDV).  Please check the whole reference list and correct where required.

Thank you for the comments. We made the changes as suggested.

Reviewer 2 Report

The authors investigated if UL12 and UL29 proteins of MDV play role(s) in the virus genome integration to the cell chromosome and maintenance of latency by utilizing recombinant MDVs that do not express these gene products and other in vitro systems. They concluded both gene products play a major role in virus replication but dispensable for the integration of virus genome and virus latency. 

The manuscript was written well and the conclusion was supported by the results. However, I have several relatively minor concerns and suggestions as below:

1. UL12 mut1 severely hampered but still allow the virus replication while mut3 completely abrogated it. This most likely implies that there is an additional translation initiation site(s) in this ORF between the mutations in mut1 and that in mut3, as far as the mutation in mut3 really abrogated the replication by affecting the protein(s) encoded in this frame. Do the authors interpret the results in this way? If the authors believe this is the case it is easier for the readership to understand the authors’ interpretation with a description regarding the relative positions of aa 136 of UL12 and any potential initiation codon(s) for UL12.5 (or other products potentially translated from 5’ to aa 136). 

2. In Fig.1 and 2 the pictures for UL12 mut2, UL12 mut3 and UL29 mut are so small that I could not see the single infected cells in these pictures. I think it is important to demonstrate the transfection was successful and the translation from the transfected BAC occurred despite the lack of replication. 

3. Page 2, lines 88. In relation to the point above, how did the authors visualize the plaques or infected/transfected cells?

4. From Page 6 line 184 -. The authors stated that they investigated the necessity of UL12 for the maintenance of latency. Why did the authors think a protein that functions in DNA damage repair plays a role in the maintenance of latency, not only for the integration? 

5. In Fig.1 legend for (A) to (D) are actually for (B) to (E), respectively. A legend for (A) is missing.

6. Page 2, line 62. “other proteins complement for the loss of these protein.” reads like the authors know UL12 and UL 29 are involved in the integration and latency and the loss can be complemented by other proteins. However, these proteins may not be involved in the integration and latency and, thus, any other proteins may not need to complement. 

7. Page 3, line 130. How much the plaque diameters were different? It is informative if the authors concisely describe the percentages in the text, not only visualize in the graph.

Author Response

Dear reviewer and editors,

We would like to thank the reviewers and editors for the evaluation of our manuscript. We have addressed the concerns of the reviewers and performed the suggested changes, which improved the manuscript compared to the previous version. You can find our direct responses to the reviewers’ comments below.

Reviewer 2 Comments & Revisions:

The manuscript was written well and the conclusion was supported by the results. However, I have several relatively minor concerns and suggestions as below:

1. UL12 mut1 severely hampered but still allow the virus replication while mut3 completely abrogated it. This most likely implies that there is an additional translation initiation site(s) in this ORF between the mutations in mut1 and that in mut3, as far as the mutation in mut3 really abrogated the replication by affecting the protein(s) encoded in this frame. Do the authors interpret the results in this way? If the authors believe this is the case it is easier for the readership to understand the authors’ interpretation with a description regarding the relative positions of aa 136 of UL12 and any potential initiation codon(s) for UL12.5 (or other products potentially translated from 5’ to aa 136).

Yes, this is also our interpretation. We included the information on the potential alternative initiation codon at aa136 that could give rise to a MDV UL12.5 orthologue in the text.

2. In Fig.1 and 2 the pictures for UL12 mut2, UL12 mut3 and UL29 mut are so small that I could not see the single infected cells in these pictures. I think it is important to demonstrate the transfection was successful and the translation from the transfected BAC occurred despite the lack of replication.

Thank you for this great suggestion. We zoomed into the images to provide a better visualization of the single infected cells. In addition, we indicated the single infected cells with arrows.

3. Page 2, lines 88. In relation to the point above, how did the authors visualize the plaques or infected/transfected cells?

We visualized the plaques using a polyclonal MDV antiserum as described previously (1). We did not include a detailed description, as this IFA staining has been described many times and is a standard procedure in the field.

4. From Page 6 line 184 -. The authors stated that they investigated the necessity of UL12 for the maintenance of latency. Why did the authors think a protein that functions in DNA damage repair plays a role in the maintenance of latency, not only for the integration?

Thank you for bringing up this point. We primarily wanted to address if UL12 and UL29 are involved in the integration process. Since integration is a prerequisite for the maintenance if the virus genome in CU91 cells and we measured genome maintenance in our assay, we felt comfortable with the title “…integration and the maintenance of the latent virus genome” to reflect our data.

5. In Fig.1 legend for (A) to (D) are actually for (B) to (E), respectively. A legend for (A) is missing.

Thank you for picking this up. We corrected the figure legend accordingly.

6. Page 2, line 62. “other proteins complement for the loss of these protein.” reads like the authors know UL12 and UL 29 are involved in the integration and latency and the loss can be complemented by other proteins. However, these proteins may not be involved in the integration and latency and, thus, any other proteins may not need to complement.

We agree with the reviewer. We changed the text to reflect both possibilities.

“… indicating that these two proteins either do not facilitate integration or that other proteins complement for the loss of these proteins.”

7. Page 3, line 130. How much the plaque diameters were different? It is informative if the authors concisely describe the percentages in the text, not only visualize in the graph.

We agree with the reviewer and included the percentages in the text.

References:

1.         Schumacher D, Tischer BK, Trapp S, Osterrieder N. 2005. The protein encoded by the US3 orthologue of Marek's disease virus is required for efficient de-envelopment of perinuclear virions and involved in actin stress fiber breakdown. J Virol 79:3987-3997.

Reviewer 3 Report

I have only a minor comment. The authos should give more emphasis on the fact herpesvirus DNA replication in total is not fully understood despite some textbooks like the "Fields" suggesting the opposite. Recent studies e.g. have shown that HSV-1 OBP is required only in the first Phase of the biphasic DNA replication and is no longer needed in the Phase when recombination driven replication occurs. It should be discussed These observations do not agree with the rolling circle replication.

Author Response

Dear reviewer and editors,

We would like to thank the reviewers and editors for the evaluation of our manuscript. We have addressed the concerns of the reviewers and performed the respective changes, which improved the manuscript compared to the previous version. You can find our direct responses to the reviewers’ comments below.

Reviewer 3 Comments & Revisions:

I have only a minor comment. The authors should give more emphasis on the fact herpesvirus DNA replication in total is not fully understood despite some textbooks like the "Fields" suggesting the opposite. Recent studies e.g. have shown that HSV-1 OBP is required only in the first Phase of the biphasic DNA replication and is no longer needed in the Phase when recombination driven replication occurs. It should be discussed These observations do not agree with the rolling circle replication.

Thank you for the suggestion. We agree with the reviewer that herpesvirus DNA replication is not yet fully understood and included this statement in the manuscript. The recent data on the biphasic nature of HSV-1 DNA replication is very exciting and an aspect that we would also like to address for MDV in the future.  

Reviewer 4 Report

Here Previdelli et al report on the role of Marek’s disease virus (MDV) UL12 and UL29 in viral lytic replication, integration into the host genome and their ability to perform DNA repair. To this end they create new viral mutants that do not synthesize the UL29 or UL12 proteins and find that mutation of UL12 diminished viral lytic replication but does not affect viral genome maintenance in a latency model.  UL29 was found to be essential for lytic replication but depletion of the protein via shRNA did not affect genome maintenance. UL12 alone increased single strand annealing but UL29 did not promote any form of DNA damage repair. The experiments are well controlled and the paper is well written aside from some typos that need correcting. I only have a couple of minor points that need addressing.

1. Can the authors show that the shRNAs function as predicted to deplete UL12 and UL29 protein or reduce RNA levels?

2. It is hard to tell from figure 1A how the different mutations in UL12 target UL12 and UL12.5. It would be nice to see the UL12.5 transcript here. Am I correct in thinking that mut1 deletes UL12 alone whereas mut2 and mut3 delete both UL12 and UL12.5? If so, is the phenotype of mut1 versus mut2 and mut3 surprising given that for HSV loss of UL12.5 function but maintenance of the function of UL12 does not have a known phenotype in lytic replication? Here it seems that loss of both proteins has a more severe phenotype that deletion of UL12 alone. Related to this, does UL12.5 in MDV have a mitochondrial localization sequence? It would be nice if the authors could expand and comment more on this. 

Minor points

The supplemental figure is not referenced in the results section. Perhaps this could be moved to the main text as an additional panel to figure 4?

Author Response

Dear reviewer and editors,

We would like to thank the reviewers and editors for the evaluation of our manuscript. We have addressed the concerns of the reviewers and performed modifications as suggested, which improved the manuscript compared to the previous version. You can find our direct responses to the reviewers’ comments below.

Reviewer 4 – Comments and Revisions:

1.         Can the authors show that the shRNAs function as predicted to deplete UL12 and UL29 protein or reduce RNA levels?

The reviewer raises an important point. To address the reviewers comment, we assessed MDV replication in CR cells that express these shRNA constructs. As we have shown in this manuscript that UL12 and UL29 are essential for lytic replication, we could demonstrate that MDV replication was severely impaired upon expression of the shRNAs. This information is now included in the text and the data is presented in Fig. S1.

2.         It is hard to tell from figure 1A how the different mutations in UL12 target UL12 and UL12.5. It would be nice to see the UL12.5 transcript here. Am I correct in thinking that mut1 deletes UL12 alone whereas mut2 and mut3 delete both UL12 and UL12.5? If so, is the phenotype of mut1 versus mut2 and mut3 surprising given that for HSV loss of UL12.5 function but maintenance of the function of UL12 does not have a known phenotype in lytic replication? Here it seems that loss of both proteins has a more severe phenotype that deletion of UL12 alone. Related to this, does UL12.5 in MDV have a mitochondrial localization sequence? It would be nice if the authors could expand and comment more on this.

Thanks for this great suggestion. As suggested by the reviewer, we included the putative UL12.5 gene in the Figure 1A. In silico predictions using TargetP 1.1 suggest that UL12.5 has a weak mitochondrial targeting peptide (mTP), however, no studies have been performed to confirm a mitochondrial localization yet.

Minor points

The supplemental figure is not referenced in the results section. Perhaps this could be moved to the main text as an additional panel to figure 4?

Thanks for picking this up. We cited the figure in the text as suggested by the reviewer.